

# Neutering of cats and dogs in Ireland; pet owner self-reported perceptions of enabling and disabling factors in the decision to neuter

Martin J. Downes[1,2], Catherine Devitt[3], Marie T. Downes[4] and Simon J. More[2]

[1] Centre for Applied Health Economics, Menzies Health Institute Queensland, Griffith University, Queensland, Australia
[2] Centre for Veterinary Epidemiology and Risk Analysis, School of Veterinary Medicine, University College Dublin, Dublin, Ireland
[3] UCD School of Geography, Planning and Environmental Policy, University College Dublin, Dublin, Ireland
[4] Greencross Vets, Brisbane, Queensland, Australia

Corresponding author
Martin J. Downes,
m.downes@epinet.net

## ABSTRACT

**Background.** Failure among pet owners to neuter their pets results in increased straying and overpopulation problems. Variations in neutering levels can be explained by cultural differences, differences in economic status in rural and urban locations, and owner perceptions about their pet. There are also differences between male and female pet owners. There is no research pertaining to Irish pet owner attitudes towards neutering their pets. This paper identified the perceptions of a sample of Irish cat and dog owners that influenced their decisions on pet neutering.

**Methods.** This study was conducted using social science (qualitative) methods, including an interview-administered survey questionnaire and focus group discussions. Data was coded and managed using Nvivo 8 qualitative data analysis software.

**Results.** Focus groups were conducted with 43 pet (cats and dogs) owners. Two major categories relating to the decision to neuter were identified: (1) enabling perceptions in the decision to neuter (subcategories were: controlling unwanted pet behaviour; positive perceptions regarding pet health and welfare outcomes; perceived owner responsibility; pet function; and the influence of veterinary advice), and (2) disabling perceptions in the decision to neuter (subcategories were: perceived financial cost of neutering; perceived adequacy of existing controls; and negative perceptions regarding pet health and welfare outcomes).

**Discussion.** Pet owner sense of responsibility and control are two central issues to the decision to neuter their pets. Understanding how pet owners feel about topics such as pet neutering, can help improve initiatives aimed at emphasising the responsibility of population control of cats and dogs.

## INTRODUCTION

Companion animal overpopulation causes significant costs to humans and governments every year (*Olson et al., 1991*; *Olson & Johnston, 1993*). Evidence suggests there is a connection between the neutering status of pets and levels of pet straying, with low levels of neutering related to higher levels of straying in pet behaviour (*Hsu, Severinghaus & Serpell, 2003*; *Diesel, Brodbelt & Laurence, 2010*). The problem of overpopulation may be attributed to numerous factors that are intertwined including; a failure among pet owners to neuter their pets (*Hsu, Severinghaus & Serpell, 2003*; *Natoli et al., 2006*; *Soto et al., 2006*; *Weng et al., 2006*), failure to implement early neutering of cats and dogs (*Ortega-Pacheco et al., 2007*; *Farnworth et al., 2013*) and poor management of stray populations (*Marston & Bennett, 2009*; *Stavisky et al., 2012*). Therefore there is a responsibility for pet owners to prevent pet pregnancies and to neuter their pets, with welfare organisations encouraging pet owners to be responsible in neutering their pets to help reduce the stray/feral dog and cat populations (*Dogs Trust, 2009*; *Dublin Society for Prevention of Cruelty to Animals, 2010*).

There are marked differences in neutering rates across the globe. These differences can be explained by variations in cultural differences and attitudes towards neutering, and differences in economic status in rural and urban locations (*Berthoud et al., 2011*; *Torres de la Riva et al., 2013*). Differences in the rate of neutering have been reported between the United States and Europe (*Trevejo, Yang & Lund, 2011*; *Torres de la Riva et al., 2013*). One US study reported the prevalence of castration at 82% in cats and 64% in dogs (*Trevejo, Yang & Lund, 2011*). In the United Kingdom, one study reported that among 431 dog owners, 54% of dogs were neutered, and there were regional differences between north and south (*Diesel, Brodbelt & Laurence, 2010*). Reported levels are similar in Hungary (*Kubinyi, Turcsán & Miklósi, 2009*), but much lower in Sweden (*Sallander et al., 2001*) and Ireland (*Downes, Canty & More, 2009*).

Perceptions owners have about their pet are also important. Owners are more likely to neuter their pet if they consider it a companion rather than a working animal (*Franti et al., 1980*; *Faver, 2009*). Increased awareness about the benefits and harms of sterilization of female cats and dogs was shown to impact positively on the decision to neuter (*Faver, 2009*). *Perrin (2009)* reported that owners of 'mostly indoor pets' believed that neutering was not necessary. Numerous reasons for not neutering have been identified using quantitative processes; including believing neutering to be unnecessary, and wanting to use the pet for breeding (*Fielding, Samuels & Mather, 2002*). Other studies showed that participants agreed that cats and dogs have the right to remain whole and have offspring (*McKay, Farnworth & Waran, 2009*). Owners thought that desexing changed the personality of their dog (*Blackshaw & Day, 1994*) and were concerned about the effect of neutering on the sexuality/masculinity of their pet (*McKay, Farnworth & Waran, 2009*). The cost of neutering also presents a barrier (*Blackshaw & Day, 1994*; *Faver, 2009*).

There are differences in neutering levels between cats and dogs, with cat owners more likely to neuter than dog owners (*Franti et al., 1980*; *Leslie et al., 1994*; *Poss & Bader, 2007*; *Downes, Canty & More, 2009*; *Faver, 2009*; *McKay, Farnworth & Waran, 2009*). Referring to neutering among pet dogs, concerns are expressed about neutering aged dogs and the

possible impact on increasing the dog's weight (*Blackshaw & Day, 1994*). There are also differences in belief and attitudes between male and female owners (*Blackshaw & Day, 1994*). Male owners equate neutering with removing the maleness of the dog, and were of the opinion that neutering can change the personality of the pet (male and female). Some 61% of male owners and 47% of female owners would not proceed with neutering their dog if they had the choice again (*Blackshaw & Day, 1994*). There are implications for the veterinary profession in the pet care recommendations it offers clients around neutering (*Scarlett, 2008*). Veterinarians can play an important role in addressing problems related to neutering and overpopulation, and counselling pet owners to take appropriate action (*Voith, 2009*). However, there are challenges to achieving the full potential of this role. *Diesel, Brodbelt & Laurence (2010)* reported that there is often variation in the veterinarian advice offered to clients. For example, there was little agreement between veterinary practices on the appropriate stage to neuter bitches, with 16.9% of practices recommending that a bitch should have a first heat before neutering, in comparison to 20.6% not recommending neutering at all (*Diesel, Brodbelt & Laurence, 2010*).

## Pet ownership and neutering in Ireland

There is no research pertaining to the opinions and perspectives of Irish pet owners towards pet neutering. This reflects the wider lack of research on pet ownership and pet care. *Downes, Canty & More (2009)* reported that some 35% of households in Ireland have one or more pet dogs, and 10.4% of households have one or more pet cats. Of these, 47.3% of pet dogs and 76.1% of pet cats were neutered. Females (in both cats and dogs) were more likely to be neutered than males (*Downes, Canty & More, 2009*). Low levels of pet neutering in Ireland, along with the uncounted number of strays reproducing, means that it is difficult to control overpopulation of cats and dogs in Ireland.

## Study objectives

Given the lack of information on pet owner perspectives on neutering in Ireland, the aim of this study is to identify the self-reported perceptions of a sample of Irish cat and dog owners that influenced their decisions on pet neutering.

## MATERIAL AND METHODS

This study was conducted using social science (qualitative) methods and reported using the EQUATOR network reporting guidelines: consolidated criteria for reporting qualitative research (COREQ) (*Tong, Sainsbury & Craig, 2007*). Qualitative methodology can provide rich and detailed information from which to develop models and theories in an area where very little research has taken place (*Barbour, 2005*), with emphasis placed on inductive inquiry and a subjective view of the insider (*Green & Thorogood, 2004*). Previous studies examining issues surrounding neutering in pets have mainly used quantitative methods such as questionnaires utilizing predetermined answers as methods of collecting data. However, this approach has some limitations (for example; these methods limit the results to the ideas of the researcher), as acknowledged by some researchers (*Robertson, 2003*). Also while statistical methods can provide some insight into the associations between

neutering and attitudes, they can rarely account for all observed variation, leaving a gap in the evidence to be explored. Focus groups are a common qualitative method. This approach is frequently used to facilitate the expression of ideas and experiences that would be left unexplored in a questionnaire or interview and clarifies participants' perspectives through debate within the group (*Kitzinger, 1995*).

## Study design

Research ethical approval was granted by the University College Dublin (UCD) Human Research Ethics Committee. Participants were required to sign a written form of consent. For the methodology, qualitative research methods—focus groups—were used. Focus groups allowed participants to openly discuss their feelings on neutering, and to indicate their own decisions around neutering their pets.

## Participant recruitment

Pet owners were recruited through six different private veterinary practices (three city practices; two in regional towns; and one in a rural area). The practices selected were a convenience sample to ensure compliance and each of these practices agreed to participation in the study. Participants were recruited by the practices, where fliers and posters were put in place and the staff was asked to highlight the research project, to encourage participants to volunteer. Participants were offered a voucher to the value of € 50 for the practice where they were recruited from. Seven focus groups were conducted with 43 participants in total; three to nine participants in each group.

## Data collection and analysis

A survey was administered prior to the commencement of the focus groups, to collect information on pet owner profile (age, location, type of dwelling, and household composition) and pet profile information (type and number of pets in participating households). Table 1 presents the participant socio-demographic profile.

An interview topic guide was used to direct all of the focus groups. Questions guided the focus groups to explore reasons for pet ownership and pet choice; views and decisions on pet neutering; feeding and weight control; and pet exercise:

- Why do you have a pet?
- Why did you choose that type of pet?
- What are your views on neutering dogs and cats?
- What influenced your decision to have your pet neutered or not?
- What are your views on pet diets, both homemade and commercial?
- What factors influence the weight of your pet?
- How do you feel about exercising your pet?

All focus groups were audio-recorded and transcribed. The coding and the analysis process were assisted using Nvivo 8 (© QSR International Pty Ltd 2007) qualitative data analysis software.

Table 1 **Socio-demographic profile for participating pet owners.** Socio-demographic profile for participating pet owners ($N = 43$).

| Socio-demographic variable | Frequency (%) | Ireland pet owners[a] % |
|---|---|---|
| **Age** | | |
| 18–24 | 3 (7.0) | 14.3 |
| 25–34 | 7 (16.3) | 24.6 |
| 35–44 | 5 (11.6) | 21.6 |
| 45–54 | 8 (18.6) | 17.1 |
| **55–64** | **14 (32.6)** | **14.15** |
| 65+ | 6 (14.0) | 8.25 |
| Total | 43 (100.0) | 100 |
| **Gender** | | |
| Female | **30 (69.8)** | **50.7** |
| Male | 13 (30.2) | 49.3 |
| Total | 43 (100.00) | 100 |
| **House type** | | |
| Apartment | 1 (2.3) | 1.4 |
| Detached | **18 (41.9)** | **57.2** |
| Semi detached | 13 (30.2) | 28.6 |
| Terraced house | 9 (20.9) | 10.0 |
| Missing | 2 (4.6) | 2.8 |
| Total | 43 (100.00) | 100 |
| **Household composition** | | |
| Lone parent with children | 3 (7.0) | 8.1 |
| Married or Cohabiting couple | 11 (25.6) | 19.84 |
| Married or Cohabiting couple with children | **13 (30.2)** | **59.3** |
| Mixed non-family household | 8 (18.6) | 4.13 |
| One person | 8 (18.6) | 8.6 |
| Total | 43 (100.0) | 100 |
| **Marital status** | | |
| Cohabitating | 3 (7.0) | 11.4 |
| Divorced or Separated | 2 (4.7) | 8.4 |
| Married | 18 (41.9) | 57.3 |
| Single | **20 (46.5)** | **21.7** |
| Total | 43 (100.0) | 98.8[b] |
| **Urban/Rural location** | | |
| Rural | 15 (34.9) | 38.1 |
| Urban | **28 (65.1)** | **61.9** |
| Total | 43 (100.0) | 100 |

**Notes.**

Bold, most frequent category.

[a] Data taken from *Downes, Canty & More (2009)*.

[b] 1.2% didn't answer.

Focus group data were grouped together using codes and themes in accordance with the technique described by *Attride-Stirling (2001)*. Minor thematic codes were developed

**Table 2 Profile of neutering among pet owners.** Profile of neutering (for cat, dogs, and both) among pet owners ($N = 43$).

| Neuter status | Cat n (%) | Dog n (%) | Both cat and dog n (%) | Total N (%) |
|---|---|---|---|---|
| Yes | 8 (29.6) | 9 (33.3) | 10 (37) | 27 (62.8) |
| Some | 1 (12.5) | 2 (25) | 5 (18.5) | 8 (18.6) |
| No | – | 5 (62.5) | 3 (37.5) | 8 (18.6) |
| Total | 9 (20.9) | 16 (37.2) | 18 (41.9) | 43 |

inductively as the transcripts were reviewed, allowing the data collected to dictate the categories for analysis. After coding, the first two authors mutually agreed on the categories that were to be used in the analysis. Two major categories related to the decision to neuter were identified: (1) Enabling perceptions in the decision to neuter (five subcategories); and (2) Disabling perceptions in the decision to neuter (three subcategories). The subcategories are as follows:

1. Enabling perceptions in the decision to neuter
   a. Controlling unwanted pet behaviour
   b. Positive perceptions regarding pet health and welfare outcomes
   c. Perceived owner responsibility
   d. Pet function
   e. The influence of veterinary advice
2. Disabling perceptions in the decision to neuter
   a. Perceived financial cost of neutering
   b. Perceived adequacy of existing controls
   c. Negative perceptions regarding pet health and welfare outcomes

## RESULTS

### Profile of neutering status

Forty three participants took part in the study. Of these, 81.4% (35) neutered at least one of their pets. For one of the focus groups, three participants did not turn up leaving only three participants available for the focus group; however this did not impact on the quality of the data collected from this focus group. Though sample sizes are small; more owners had neutered cats than dogs, relative to the sample size. Eight pet owners neutered some of their pets, and the same number did not neuter their pets. Table 2 details the profile of neutering among pet owners in the study.

### Enabling perceptions in the decision to neuter
#### *Controlling unwanted pet behaviour*
For pet owners, that neutered their pets, neutering provided a means of controlling pet behaviour and reducing the propensity for unwanted and undesired behaviours for the pet
owner. Animal behaviours that were identified as unfavourable included fighting between pets, and straying. Neutering reduces the attraction of other cats and dogs to the pet owners' home, and prevents unwanted pets.

> '*It's a case of health issues and trying to keep the cats out of fights.... I think if they're not neutered, they want to be out more. Especially at night and that's putting them at risk from the traffic*'

> '*My dog is neutered... he's a cocker spaniel and they have a reputation for being hyper. Neutering will calm him down. I don't know what happens when dogs go into heat but I do know that the males go mad so I just thought it would be safer, as we walk him [with]out the lead. I'd be petrified if he ran off. I wouldn't know what to do, so I would agree with neutering*'.

> '*I think with cats, you want them there, and a neutered cat stays around the house, they don't wander*'.

> '*One tabby was neutered when I got it and I decided to neuter the others because they would mark their territory and probably fight more. So they are all neutered.*'

### Positive perceptions regarding pet health and welfare outcomes

Much discussion was had on the health consequences of neutering for pets. Pet owners referred to the beliefs of others, and their own:

> '*People said he'd* [pet dog] *be sluggish, he'd be lethargic, and he'll put on weight. I never saw any change. He was a young happy dog. There are these myths going around that [neutering] will change your dog's character. I've never seen that*'.

Pet owners, in favour of neutering, regard neutering as an effective way of ensuring good animal health for their pets. In addition to controlling the pet's behaviour and reducing the propensity for unwanted pet behaviours, neutering pets was seen as a way of reducing the risk of the spread of disease, infections, and harm caused by fighting (and mating) between animals. For these reasons, neutering was seen as a way of prolonging the life span of the pet.

> '*With the cats, it's a case of health issues, to avoid the risk of Feline AIDS. They can pick up so much if they're out and fighting*'.

> '*A male cat, I had him neutered because I didn't want him to catch feline AIDS*'

> '*It will prevent them* [cat] *having infection or uterine cancer... or mammary cancer*'.

Both cat and dog owners refer to neutering as increasing the life span of the pet.

> '*If you have the dog neutered, the bitch neutered, it can extend her life because they don't have to go through the ordeal of giving birth, pups. That can actually add another year or two to the bitch's life span, so that's why I got my present dog neutered*'.

> '*Neutering prolongs the males*' [cat] *life. They're not fighting and spreading disease*'.

*'They are pets, it can increase their lifespan because of cancer and diseases, I wanted them* [dogs] *to live a couple of years longer, it may be selfish, they may have had some great experiences, but I'll hang onto them as long as possible'.*

### Perceived owner responsibility

Owner sense of responsibility was apparent in the statements of pet owners who are in favour of neutering. Owners felt a responsibility to reduce unwanted pregnancies, and prevent over-population of unwanted cats and dogs.

*'A dog yes, you don't want to be responsible for your pet creating a litter of pups or kittens'.*

*'Every time you hear figures, how many pets—dogs and cats—that have to be put down every year, because they can't be kept, the shelters are all overrun with them. It's just the thought of it going on, is just horrible'.*

*'I don't want the responsibility of having kittens or having to find homes. So it is the responsible thing to do'.*

*'Neutering is more responsible and there are too many puppies around. I have cats and I let them outside. I would hate to have it on my conscience that they were the cause of some other cat having a litter'.*

Pet owner comments reflect an emotional perspective on the problem of overpopulation, and not wanting to deal with the implications of finding homes for unwanted kittens and pups, or the implications at an emotional level for the owner.

### Pet function

Keeping a pet (dogs were specifically mentioned in this study) for breeding purposes was identified as a reason for deciding not to neuter.

*'*[Dogs] *should be neutered unless there is a good reason for breeding from them'.*

*'I had my cat neutered and I can see no point not to, unless you particularly want to breed from the animal for some reason'.*

Only one pet owner indicated that they were breeding from their pet dog, and therefore, decided against neutering. There was no reference made to other functional related reasons for owning a pet, e.g., working animal, companionship, etc.

### The influence of veterinary advice

Only four pet owners referred specifically to the role of veterinary advice in informing their decision to neuter. There was general consensus among the groups that neutering would be complied with if medically required. Two of these pet owners noted they were not in favour of neutering, but complied with medical advice. There were mixed opinions on this decision, with reference being made to a loss of perceived control over the decision:

*'So the advice was that medically I should do it* [neuter the pet dog], *so I did it and I didn't really think about the rights and wrongs of it at all really'.*

'*The vet would make the decision for us; she did say the female was quite small to have pups*'.

'*My decision ended up having to be taken from me… the real decision was that she then got a false pregnancy and the vet said to me this can be a precursor of cancer type of thing and really I'd be better off doing it*'.

The influence of media featured very little in the focus groups, but pet owners made reference to information on the number of injured, unhealthy and euthanized cats and dogs.

## Disabling perceptions in the decision to neuter

### Perceived financial cost of neutering

Financial cost was identified as a barrier to improving the prevalence of neutering of pet cats and dogs. This barrier was identified by five participants; though all five had their pet neutered. Instead, concern was raised that the financial costs of neutering would prevent others from neutering their pets.

'*I think the cost in Ireland is extremely high…. I had my two dogs done at the same time eight years ago and it was about € 350 to get them neutered by the vet*'.

'*For the two* [cats]*… that was my bill when I went to pick them up. That's an awful lot of money…there are people who genuinely can't afford it…*'

### Perceived adequacy of existing controls

There was an overwhelming perception among those who did not have their pets neutered that adequate control measures were in place, or that neutering was not necessary because the pet was always indoors, or within sight of the owner. These measures include keeping the pet inside a controlled environment, such as the owners' house.

'*The dog we have at the moment is not neutered. It depends on the dog and the environment which it lives, whether you have a garden, whether other people are home during the day, whether the dog is taken out on the lead only*'.

'*When we got the dog, she was not neutered simply because she was always under control and there was no one living near us. So all of us made sure she was tied, up in her pen*'.

'*No, he's* [pet dog] *not neutered. He's around us all the time. He's under strict control around the house*'.

'*One pet which is completely indoors—She's a total house dog. Someone's always with her if she's outside, there's no need for her to be neutered*'.

'*[Neutered dogs] get too fat and lazy and it's not hard to lock up a bitch for a month twice a year. I have dogs and bitches at home and I can cope with it…. if you've a bitch in heat, you lock her up. I don't agree* [with] *neutering*'.

Specific reference was made to dogs; dogs were perceived as easier to control than cats.

> '*I didn't neuter the dog but he was never loose outside. There was never any chance he was going to get himself into trouble, because he was either inside with us or outside with one of us, but I did neuter both of the cats. I did that because I didn't want the male to get himself in to trouble in other peoples gardens*'.

> '*I can understand with cats* [the need to neuter] *because they're out wandering and stuff, but with a dog and you know where they are all the time*.'

In this instance, the cats and a male dog are neutered. However, the decision was made not to neuter the female dog:

> '*All my cats are neutered and only one male is neutered… cats get diseases when they're out and around whereas the dogs are more home birds*'.

### Negative perceptions regarding pet health and welfare outcomes

As with the decision to neuter, concerns pertaining to animal health were also influential in the decision not to neuter. These concerns reflect pet owners' belief regarding the consequences and outcomes of neutering.

> '*When you get them* [cats] *neutered, they are inclined to put on a lot of weight, and they lose their shape*'.

> '*It is nice to leave them* [pets] *and not play around with them too much… just leave them as their natural self*'.

Statements point to the belief that neutering is unnatural for the pet. Among some owners of neutered bitches, concern was expressed about the invasiveness of the procedure, how sick it had made their pet, and contributed to weight gain. Given this experience, these owners expressed reluctance to neuter future pets.

## DISCUSSION

### Overview

In this study, the majority of the participant group had neutered all of their pets (62.8%).

Self-reported perceptions were organised into those that were (i) enabling (i.e., supported the decision to neuter) and (ii) disabling (i.e., were against the decision to neuter). All pet owners in favour of neutering had neutered their pets. A minority of those against neutering also had their pets neutered, in compliance with medical advice. Enabling perceptions that supported the decision to neuter included: a desire to control unwanted behaviours (such as straying and fighting); concerns over animal health; a perceived sense of owner responsibility; pet function; and because of veterinary advice. Disabling perceptions that influenced the decision not to neuter included: the perceived financial cost of neutering; the adequacy of existing controls; and concerns over animal health. It is hoped that, in addition to encouraging further research for an Irish context, the results in this paper will contribute to a better understanding of pet owner behaviour, and contribute to informing veterinary advice and support for adequate pet care.

## Discussion of key findings

The health benefits of neutering for pets included decreased risk of some cancers, and increased longevity (*Smith, 2014*). In this study, there was a clear connection between the desire to control pet behaviours (such as straying and fighting), perceived perceptions regarding pet health and welfare outcomes, and the objective of preventing inconvenient implications for the pet owner (such as dealing with unwanted kittens). Neutering was described as prolonging the life span of the pet. This may suggest that owners' decisions are influenced by information beyond their own experiences, such as from a veterinarian (*Faver, 2009*); however, explicit reference to veterinary advice was made by only a small number of pet owners. For those in favour, neutering was generally considered a matter of responsibility, with reference being made to the need for cat and dog population control. This suggests a level of awareness among these pet owners not only of the health related benefits of neutering, but also the wider problems associated with overpopulation.

The importance of normative beliefs and perceived ability are important in explaining the relationship between responsibility and behaviour among pet owners (*Rohlf et al., 2010*). Recent welfare organizations marketing strategies emphasise the responsibility of population control on pet owners (*Dogs Trust, 2009*; *Dublin Society for Prevention of Cruelty to Animals, 2010*). The results show that responsibility and control are two central issues. Pet owners in favour of neutering commented on their sense of responsibility—reflecting an emotional component and an awareness of the implications for the wider cat and dog population if they did not neuter their pets. These pet owners also talked about not wanting to have to respond to unwanted offspring. This reflects their sense of control and responsibility over their pet's behaviour. Neutering provides a means of controlling this behaviour and emotionally reassuring the pet owner. Those against neutering emphasised the adequacy of existing control measures—suggesting a high level of perceived control over the behaviour of their pet, the pet's environment, and the owner's own ability to keep the pet under observation. This is similar to the findings of *Perrin (2009)* who reported that owners of 'indoor' pets believed that neutering was not necessary. Owner responsibility was not mentioned by those not in favour of neutering—though pet-related health concerns were emphasised—with pet owners expressing concerns about the invasiveness of the operation, and the risk of pet obesity. Differences were recorded in opinions towards neutering of cats and dogs. Cats were regarded as wanderers, less easy to control and more prone to picking up infection and disease. Dogs were seen as easier to control, and therefore, control measures (such as keeping the dog in a controlled environment, such as indoors) were regarded by some as adequate.

Though financial concerns did not feature strongly in the results, the research literature does show that the cost of neutering can present a barrier to pet owners (*Blackshaw & Day, 1994*; *Faver, 2009*). At the time of research, the economic climate in Ireland has resulted in less disposable income for people. Pet owners may regard neutering as less of a priority, and instead, implement measures to keep the pet indoors. Recent media coverage has highlighted a growing problem of dog and cat abandonment, associated with a weak economic climate (for example, an article in The Irish Times '*Overcrowded animal centre in urgent*

*appeal for 'responsible' owners after rise in abandoned pet'* (*The Irish Times, 2012*) and an article in the Irish Examiner '*Abandonment on rise during recession'* (*Irish Examiner, 2012*).

As there are varying attitudes to neutering in different settings, it is important to conduct research around this area in different countries. Even the level of neutering varies across different countries, for example; only 3% of dogs in the Coquimbo region of Chile (*Acosta-Jamett et al., 2010*) and 1.1% of dogs in Sweden (*Sallander et al., 2001*) were neutered compared to 47.3% of pet dogs in Ireland (*Downes, Canty & More, 2009*). There are even within country variations, depending on the methodology used to collect the data i.e., in the US, 64% of dogs are neutered according to *Trevejo, Yang & Lund (2011)* compared to 83% as estimated by the American Society for the Prevention of Cruelty to Animals (*ASPCA, 2011*). As far as the authors are aware, the current study is the first study to use qualitative methods to provide rich data on the attitudes and behaviours of pet owners towards neutering and is the first to identify factors associated with neutering in Ireland. With this in mind the data will benefit current practice and to aid in the development of future research in Ireland.

## Implications for veterinary advice on neutering

Understanding how pet owners feel about topics such as neutering gives veterinary services the knowledge and understanding to improve their relationship and communication with pet owners (*Perrin, 2009*). It has been suggested that veterinary practitioners need to communicate more effectively with pet owners around the time of neutering, to ease the burden of neutering on the pet and the owner and to encourage owners to neuter future pets (*Blackshaw & Day, 1994*). *Diesel, Brodbelt & Laurence (2010)* however, found variations between veterinarian recommendations to pet owners on neutering. This may suggest a lack of consistency in approach and gaps in information on best practice, within the veterinary profession. Though, in this study, the veterinarian did not feature strongly in perceptions of neutering, there is value in considering the role that veterinarians play in working with pet owners. Often the decision to neuter was made by the veterinarian, and some pet owners spoke of the decision being made for them. Effective communication is central. *Coe, Adams & Bonnett (2008)* emphasises the importance of educating clients, providing choices, and using two-way communication. These are important factors that need to be considered by veterinarians when advising on neutering.

Various initiatives have been launched in Ireland. The Irish Society for the Prevention of Cruelty to Animals (ISPCA) issued an information leaflet '*It pays to spay or neuter your pet*' (Irish Society for Prevention of Cruelty to Animals) which presents information on neutering, and clarification around neutering myths. Another example—the Dogs Trust launched the "it's nicer to neuter" campaign (*Dogs Trust, 2009*), in an effort to reduce the number of unwanted dogs that are euthanized annually. However, behavioural change cannot be attributed to information alone; attitudinal changes are also required (*Ajzen, 1991*). The promotion of owner responsibility within the wider community (outside of the confines of the owner's home) is one area that can be targeted by neutering initiatives. In addition, the results clearly suggest that pet health is important for pet owners (those for

and against neutering). Given the centrality of health concerns for pet owners, attempts to promote neutering should take into account the role of veterinary support and advice in adequately informing pet owners on the health benefits of neutering (*Faver, 2009*).

### Limitations in the study design and recommendations for future research

In this sample there was an over representation of female owners. There is evidence in the research that shows differences in belief and attitudes between male and female pet owners, with male owners expressing concern over a change to the pet's personality as a result of neutering (*Blackshaw & Day, 1994*). It may be argued that future research pertaining to an Irish population should seek to differentiate differences in belief and levels of neutering between male and female owners. The sample was not stratified by socio-economic group, though different geographical locations, urban and rural, were chosen to minimise this bias. The recruitment of pet owners took place through private veterinary practices. Therefore, it is probable to suggest that participants were more engaged in their pet's health, and could afford to avail of veterinary health care services.

While the sample size is not powered to conduct statistical hypothesis testing, as it is not appropriate in a qualitative setting, rather information was collected until saturation of the data was reached. In qualitative research; sampling is conducted to select the most appropriate participants, in this case pet owners (*Daly et al., 2007*), therefore it may not be necessary to conduct random sampling techniques. In addition, focus groups provide a platform to develop an understanding of participants' behaviour and attitudes based on their knowledge of the situation being addressed and provide a more pleasing setting for discussion to occur leading to richer and more realistic data (*Heary & Hennessy, 2002*; *Green & Thorogood, 2004*). Thematic analysis allows an interpretation of the data based on experience, and educational and social interactions leading to a better understanding of the issue being addressed (*Green & Thorogood, 2004*). Also, to reduce bias, this study was conducted and the data is presented and interpreted in a way that meets the critical appraisal guidelines as outlined by the Cochrane Collaboration Qualitative Methods Group to ensure the high quality of the study (*Harris, 2011*).

## ACKNOWLEDGEMENTS

The authors would like to acknowledge the invaluable input from all pet owners who participated in the study and Barna Veterinary Clinic, Blessington Pet Hospital, Primrose Hill Veterinary Hospital, Raheny Veterinary Hospital, Sandymount Pet Hospital and The Animal Health Centre for participating in the study. The authors would also like to thank Margret Nolan and Net Doyle for transcribing the focus group recordings.

### Funding

Funding was provided by University College Dublin. The funders had no role in study design, data collection and analysis, decision to publish, or preparation of the manuscript.

## Grant Disclosures

The following grant information was disclosed by the authors:
University College Dublin.

## Competing Interests

Marie T. Downes is an employee of Greencross Vets, QLD, Australia. Catherine Devitt was a self employed Consultant Social Science Research Professional located at Glendalough, Co. Wicklow, Ireland.

## Author Contributions

- Martin J. Downes conceived and designed the experiments, performed the experiments, analyzed the data, contributed reagents/materials/analysis tools, wrote the paper, prepared figures and/or tables, reviewed drafts of the paper.
- Catherine Devitt analyzed the data, wrote the paper, prepared figures and/or tables, reviewed drafts of the paper.
- Marie T. Downes performed the experiments, analyzed the data, wrote the paper, reviewed drafts of the paper.
- Simon J. More conceived and designed the experiments, contributed reagents/materials/analysis tools, wrote the paper, reviewed drafts of the paper.

## Human Ethics

The following information was supplied relating to ethical approvals (i.e., approving body and any reference numbers):

Research ethical approval was granted by the University College Dublin, Human Research Ethics Committee.

## Supplemental Information

Supplemental information for this article can be found online at http://dx.doi.org/10.7717/peerj.1196#supplemental-information.

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
