# Peer review of "Neutering of cats and dogs in Ireland; pet owner self-reported perceptions of enabling and disabling factors in the decision to neuter"

_PeerJ, doi:10.7717/peerj.1196_

## Round 0.1 · original submission · Major Revisions

The manuscript has to be modified and improved according to the reviewer comments before being accepted for publication.

Reviewer 1 ·

Basic reporting

The paper brings an important issue and less studied in order to provide valid parameters in the international literature. However, despite having a good literature review in the background, it does not dialogue with the text. For example, the chosen population has a different profile of the general population regarding the proportion of people who neutering their pets (the general population of Ireland is around 50-60% and the study population more than 80%). Considering that this is the focus of the study, the choice of this population may have brought a bias to the results obtained. The authors could be used this information, however, there is no connection of this fact to the results of the study.
Thus, although the study is about relevant theme, the way it was written does not allow their results to be used properly.

Experimental design

The methodology was interesting and innovative when in the analysis of qualitative data.
However, the chosen population may have introduced a bias to the results obtained.
This is a population with a specific profile and well defined (men, single, 55-64 years). It is not clear in the text if there was a sample calculation to determine that 43 individuals would be sufficient and/or what were the criteria for choosing this population, only that come from private veterinary practices. The choice should been randomly joined in the practices, after calculation of a representative sample number, for example.
Moreover, no less serious, this is a population in which more 80% of pets are neutered. The objective of the study was to identify perceptions of owners of dogs and cats from Ireland who influenced their decision on pet neutering. To achieve this, the study population should be from people who decided to neuter and decided not to neuter in equal numbers. Once the most of population studied neutered their pets, may also have been introduced a bias to the results obtained.
Finally, despite the analysis of the data have been interesting, the choice of population to be studied was not adequate to achieve the proposed objective.

Validity of the findings

Similarly to the study designs, the validity of the results obtained in the study are compromised.
The choice of the study population may have introduced two types of selection bias to the results obtained (as already detailed under "experimental design").
Moreover, even if the study population had been calculated and a n representative random sample, the fact that they are from private veterinary practices, causes it to produce specific and localized results. Ie they can not be extrapolated to the population of the city or even in Ireland.
Therefore, the results obtained are not valid and has little impact on the international literature.

Additional comments

The study is a relevant subject, with few estimates of valid parameters in the international literature. However, for it to reach the objective and bring something new and with impact, should be redesigned and rewritten. Not recommend it for publication.

·

Basic reporting

The authors note on Line 86 that the literature is devoid of scientific research “pertaining to the opinions and perspectives of Irish pet owners towards pet neutering.” This makes the topic of this manuscript welcome, for one should no more consider that solutions that are relevant to other countries in Europe are relevant to Ireland than solutions that are relevant in, say, the United States or Australia.

Lines 65-66: Apparently for many people the reasons for not neutering go well beyond the two cited. Some people believe it is physiologically advantageous to not neuter; some people believe that female pets are more fulfilled if they have at least one litter; others want family members (particularly children) to observe birthing (apart from using a pet for breeding purposes). This could be elaborated upon based on findings from other countries before discussing the findings in the geographical context of this manuscript.

Experimental design

Of course, the value of such research is predicated on the participants representing the source population of pet owners who may, or may not, use veterinary care (i.e., is the study population valid?). In addition, a study size must be robust enough to allow reasonably precise estimates (i.e., are the outcome statistics reasonably precise?). This will highly influence the gravity and acceptance that readers will assign to the findings and conclusions of this manuscript. In an ideal world, a well-endowed funding agency would negate any financial limitations of such work; in reality, the authors were probability limited by the voucher costs that were apparently intramurally provided (Lines 391-392) (as well as participation rates, etc.). Still, the restriction of the study population to people using veterinary practices leads to questions about representativeness, because people who utilize veterinarians for care are undoubtedly more likely to surgically neuter their pets than people who forgo such care. This is borne out by the finding (Line 149) that the neutering proportion (81%) was higher in the sample than those reported in Lines 89-90. In addition, the sample was described as one of “convenience,” which suggests that no particular sample size was sought in advance to achieve a certain level of precision. And even while 35 individuals who neutered at least one of their pets is not a large number for descriptive purposes, the true limiting factor is in the remaining eight individuals who did not neuter at least one of their pets: it is troubling to make any conclusions about such individuals or attitudes based on this small sample size. These issues/limitations deserve additional elaboration (e.g., in addition to Lines 381-383) in the Discussion.

Validity of the findings

Line 157: When the authors write “For pet owners, …”, are they referring to all pet owners in the study who neutered at least one of their pets?

Table 1: Is there demographic information about Ireland available to see how different the study population was from the country in its entirety? I’m not advocating statistical testing (the numbers in the strata are too small), but I think it could provide an interesting perspective about the representativeness issue raised above.

Table 2: Here I would welcome seeing confidence intervals for the proportions, so readers can appreciate how precise or imprecise these proportions are. For example, the authors present a seemingly precise proportion of 12.5% based on a numerator of only one owner with a cat. This will be quite simple yet useful to provide.

I had difficulty reconciling the following two statements: “Of these [forty three participants], 81.4% (35) neutered at least one of their pets” (Lines 149-150) with “In this study, a significant proportion of the participant group had neutered their pets (62.8%)” (Line 290). I also assume that the use of the word “significant” is not in a statistical hypothesis testing sense.

Additional comments

Just as different countries have their own culture(s), so do their pet owners with respect to attitudes towards and implementation of surgical neutering (male castration, female spaying). Because broad solutions will not necessarily apply to every country (each with its own unique pet and human populations with attributes), it becomes obviously necessary to define the extent of the “problem” (i.e., not neutering pets) and then understand determinants that contribute to the “problem.” Clearly, the authors contend that such a “problem” exists: “Low levels of pet neutering in Ireland … means that it is difficult to control overpopulation of cats and dogs in Ireland” (Lines 92-93). This largely qualitative sociologic study is apparently the first attempt in this country to contribute to an understanding of these issues. In a country where such issues are well-studied, quantitative studies have predominated. This was not possible given the scope of this research, and so represents a modest but still novel contribution, and will perhaps lead to more robust work in Ireland in the future.

---

## Round 0.2 · Minor Revisions

Please consider the suggestions of Reviewer 2 in a revised version.

·

Basic reporting

No comments.

Experimental design

See General Comments for Authors.

Validity of the findings

See General Comments for Authors.

Additional comments

PeerJ 2015:01:3736:1:1:REVIEW

In reviewing this manuscript’s revisions, and the comments of all reviewers, it is apparent that the latter's expectations for this manuscript are not entirely concordant with those of the authors. While this does not disqualify the manuscript, it is incumbent on the authors to clarify to their readers what the scope of this research is.

Consider the following sentence from the Abstract: “Given the lack of information on pet owner perspectives on neutering in Ireland, the aim of this study is to identify the self-reported perceptions of Irish cat and dog owners that influenced their decisions on pet neutering.” One is thus left to believe that the target population for external validity purposes is literally “Irish cat and dog owners.” As the authors note in their rejoinder, this is not the case. Therefore, the Abstract should be revised to avoid any confusion early on that could lead reviewers, if not readers, to reach the conclusion that the goals of the study were not attained.

In a similar vein, the authors write that: “This study was conducted using social science (qualitative) methods.” Fair enough, but most epidemiologists have been raised in a hypothesis generation/hypothesis testing/quantitative environment, so a departure from this entrenched scientific method deserves more elaboration and justification if the work is to gain acceptance. Perhaps there are specific meanings and methods that correspond to “social science” methods, but many in the authors’ target audience will not know what they are. I urge the authors to therefore provide a scientific – albeit qualitative – justification for their approach to this research.

---

## Round 0.3 · accepted · Accept

It is my pleasure to accept your manuscript.